# ARTIFACTGEN: WGAN-GP AND DIFFUSION FOR LABEL-AWARE EEG ARTIFACT SYNTHESIS

## ABSTRACT

Artifacts in electroencephalography (EEG)—muscle, eye movement, electrode, chewing, and shiver—confound automated analysis yet are costly to label at scale. We study whether modern generative models can synthesize realistic, label-aware artifact segments suitable for augmentation and stress-testing. Using the TUH EEG Artifact (TUAR) corpus, we curate subject-wise splits and fixed-length multi-channel windows (e.g., 250 samples) with preprocessing tailored to each model (per-window min–max for adversarial training; per-recording/channel $z$-score for diffusion). We compare a conditional WGAN-GP with a projection discriminator to a 1D denoising diffusion model with classifier-free guidance, and evaluate along three axes: (i) fidelity via Welch band-power deltas ($\Delta\delta$, $\Delta\theta$, $\Delta\alpha$, $\Delta\beta$), channel-covariance Frobenius distance, autocorrelation $L_2$, and distributional metrics (MMD/PRD); (ii) specificity via class-conditional recovery with lightweight $k$NN/classifiers; and (iii) utility via augmentation effects on artifact recognition. In our setting, WGAN-GP achieves closer spectral alignment and lower MMD to real data, while both models exhibit weak class-conditional recovery, limiting immediate augmentation gains and revealing opportunities for stronger conditioning and coverage. We release a reproducible pipeline—data manifests, training configurations, and evaluation scripts—to establish a baseline for EEG artifact synthesis and to surface actionable failure modes for future work.

## 1 INTRODUCTION

Artifacts in electroencephalography (EEG)—including muscle activity, eye movements, electrode noise, chewing, and shivering—routinely confound automated analysis and downstream clinical applications by distorting morphology, spectra, and cross-channel correlations. While artifact removal is well studied (Urigüen & García-Zapirain, 2015; Jiang et al., 2019), realistic *synthesis* of artifact segments can complement curation efforts by enabling data augmentation, algorithm stress testing, and robustness benchmarking without additional human labeling. The challenge is to synthesize multi-channel windows that remain label-aware while respecting signal morphology, spectral structure, and channel covariance.

We introduce ARTIFACTGEN, a practical and *reproducible* framework for artifact-conditioned EEG synthesis built on subject-wise splits from the TUH EEG corpus and its artifact-annotated subset (TUAR) (Hamid et al., 2020;?). ARTIFACTGEN marries two complementary generative paradigms: (i) a conditional WGAN-GP with a projection discriminator for stable, label-aware synthesis (Gulrajani et al., 2017; Miyato & Koyama, 2018), and (ii) a denoising diffusion model using a 1D U-Net with FiLM-style conditioning (Perez et al., 2018) and classifier-free guidance for controllability and sample quality (Ho et al., 2020; Ho & Salimans, 2022). The pipeline standardizes preprocessing for fixed-length windows with configurable normalization, exposes training/evaluation via YAML configs, and ships analysis notebooks to facilitate faithful ablations and apples-to-apples comparisons.

Beyond single-number heuristics, ARTIFACTGEN emphasizes a time-series-appropriate evaluation suite: (i) signal-level descriptors (e.g., Welch band-power deltas and covariance/ACF distances) to test morphology and spectra (Welch, 1967); (ii) feature-space metrics (FID/KID/PRD) to quantify fidelity–coverage trade-offs (Heusel et al., 2017; Binkowski et al., 2018; Sajjadi et al., 2018); and (iii) functional tests—train-real/test-synth, train-synth/test-real, and AugMix-style augmentation—to probe utility and robustness (Hendrycks et al., 2020). We release code, configuration files, and

notebooks to support rigorous baselining and community progress on EEG artifact generation and augmentation.

## 1.1 CONTRIBUTIONS

- A subject-wise pipeline to curate labeled artifact windows with robust normalization and fixed-length padding/truncation (Obeid & Picone, 2016; Hamid et al., 2020).
- A conditional WGAN-GP with projection discriminator for stable, label-aware synthesis (Gulrajani et al., 2017; Miyato & Koyama, 2018).
- A 1D diffusion model with FiLM conditioning and classifier-free guidance (Perez et al., 2018; Ho et al., 2020; Ho & Salimans, 2022).
- A transparent evaluation suite spanning signal-level properties, feature-space distances (FID/KID/PRD), and functional tests including AugMix-style augmentation (Heusel et al., 2017; Binkowski et al., 2018; Sajjadi et al., 2018; Hendrycks et al., 2020; Welch, 1967).
- A compact, quantitative comparison with tabulated bandpower errors (Table 1), channel-wise effects (Table 2), and MMD (Table 3).
- Embedding-space analyses (t-SNE/UMAP; Fig. 2) to cross-check alignment between metrics and structural similarity.

## 2 BACKGROUND

Electroencephalography (EEG) is indispensable in clinical neurophysiology, yet real-world recordings are rife with non-neural artifacts—ocular movements, muscle activity, chewing, shivering, and electrode noise—that degrade downstream analysis and confound learning systems. Decades of signal-processing work have characterized these artifacts and proposed removal strategies, underscoring their broad spectral footprint and nonstationary morphology (Urigüen & García-Zapirain, 2015). Large public corpora such as the Temple University Hospital EEG (TUH EEG) data (Obeid & Picone, 2016) and its artifact-focused subset, the Temple University Artifact Corpus (TUAR) (Hamid et al., 2020), enable supervised benchmarking but remain label- and condition-limited for training robust models that must generalize across subjects, montages, and acquisition conditions.

Generative modeling offers a complementary route: synthesize realistic artifact segments to (i) augment scarce classes, (ii) stress-test detector robustness, and (iii) study failure modes under controlled perturbations. Among competing paradigms, Generative Adversarial Networks (GANs) and Denoising Diffusion Probabilistic Models (DDPMs) dominate recent progress. GANs are sample-efficient but historically fragile; Wasserstein GANs with gradient penalty (WGAN-GP) improved stability and convergence by enforcing a soft Lipschitz constraint on the critic (Gulrajani et al., 2017). For class-conditional generation, the projection discriminator embeds labels into the critic, providing a principled, label-aware training signal that scales well with many classes (Miyato & Koyama, 2018).

Diffusion models take an alternative path, learning to invert a gradual noising process and achieving state-of-the-art generative quality across domains (Ho et al., 2020). Practical refinements—including learned variance and hybrid training objectives—further reduce sampling cost while preserving fidelity (Nichol & Dhariwal, 2021). For conditional synthesis, classifier-free guidance yields strong label adherence with tunable trade-offs between diversity and faithfulness (Ho & Salimans, 2022). Compared to GANs, diffusion models typically exhibit more stable training and better mode coverage, albeit with higher sampling latency—an important consideration for time-series pipelines.

Evaluating synthetic EEG requires metrics aligned with neurophysiological structure rather than image heuristics. Power spectral density (PSD) via Welch's method provides band-power comparisons in canonical $\delta/\theta/\alpha/\beta$ bands, capturing key frequency-domain shifts induced by artifacts (Welch, 1967). Temporal structure can be probed by autocorrelation statistics, while cross-channel dependencies—crucial in multi-lead EEG—are reflected in covariance distances. Complementary distributional tests quantify fidelity and coverage: precision–recall for distributions (PRD) disentangles sample quality from support coverage (Sajjadi et al., 2018), and maximum mean discrepancy (MMD) offers a kernel-based two-sample statistic sensitive to higher-order differences (Binkowski et al., 2018).

Within EEG specifically, recent surveys document growing use of GANs for augmentation and domain shifts across BCI and clinical tasks, while highlighting persistent gaps in label control,

spectral realism, and reproducibility (Habashi et al., 2023). In parallel, compact discriminative backbones (e.g., EEGNet) provide downstream validators whose behavior on synthetic vs. real segments can reveal class-specific mismatches (Lawhern et al., 2018). Together, these developments motivate a careful, *label-aware* comparison between conditional WGAN-GP and conditional diffusion for artifact synthesis on TUAR, under subject-wise splits and an evaluation suite that balances spectral, temporal, multichannel, and distributional criteria.

## 3 RELATED WORK

**Wasserstein GANs and conditioning for time series.** The Wasserstein GAN with gradient penalty (WGAN-GP) stabilizes adversarial training by softly enforcing the 1-Lipschitz constraint (Gulrajani et al., 2017). For semantic control, class-conditional GANs with a projection discriminator inject labels into the critic, improving fidelity and label-faithfulness without auxiliary classifiers (Miyato & Koyama, 2018). In 1D signals (audio and other biosignals), fully convolutional generators and discriminators (e.g., WaveGAN) motivated architectural choices that preserve local stationarity while capturing long-range context (Donahue et al., 2019).

**Diffusion models for (neuro)physiological time series.** Denoising diffusion probabilistic models (DDPMs) learn to invert a fixed noising process and now set the bar for sample quality across domains (Ho et al., 2020; Nichol & Dhariwal, 2021; Dhariwal & Nichol, 2021). Classifier-free guidance (CFG) trades off diversity and fidelity without a separate classifier, a practical tool for label-aware synthesis (Ho & Salimans, 2022). Although most diffusion results target images, several works adapt them to time series via 1D U-Nets and score-based objectives; e.g., DiffWave for raw waveform synthesis and broader score-based SDE frameworks for sequences (Kong et al., 2020; Song et al., 2020a). In neurophysiology specifically, recent models generate multichannel EEG/ECoG with strong realism and controllability (Vetter et al., 2024; Tosato et al., 2023). Our 1D U-Net with label conditioning follows this line, emphasizing artifact-aware control for EEG. We adopt FiLM-style conditioning to modulate intermediate features by condition vectors (Perez et al., 2018), and a U-Net backbone (Ronneberger et al., 2015) tailored to 1D signals.

**EEG datasets and artifact corpora.** We build on the Temple University Hospital EEG (TUH-EEG) ecosystem, the largest open clinical EEG collection (Obeid & Picone, 2016). For artifact-centric synthesis and evaluation, the TUH EEG Artifact Corpus (TUAR) provides dense annotations for common artifacts—eye movements, muscle, chewing, shiver, and electrode events—enabling subject-wise splits and label-aware benchmarking (Hamid et al., 2020).

**Evaluation of generative models for EEG.** Image-native quality metrics such as FID (Heusel et al., 2017) and KID (polynomial-kernel MMD) (Binkowski et al., 2018), and distributional precision/recall curves (Sajjadi et al., 2018; Kynkäänniemi et al., 2019) rely on features from a pretrained encoder; for EEG, we analogously extract features from artifact classifiers (e.g., EEGNet-style encoders) to adapt these ideas (Lawhern et al., 2018). In addition, *two-sample testing* provides principled sample–realism checks: kernel MMD (Gretton et al., 2012) and classifier two-sample tests (C2ST), where a held-out accuracy near chance indicates good sample quality (Lopez-Paz & Oquab, 2017). Domain-aware signal metrics complement feature-space tests: Welch band-power deltas in canonical bands (Welch, 1967), channel-covariance Frobenius distances, and ACF-based distances probe spectral shape, spatial coupling, and temporal dependence, respectively. We also report a simple 1-NN accuracy in the learned feature space as a pragmatic C2ST variant.

**Utility as the ultimate yardstick.** Beyond proxy metrics, *functional* evaluation—training down-stream models with synthesized data—best captures whether synthetic artifacts help real tasks. Time-series work has advocated train-on-synthetic, test-on-real (TSTR) to quantify downstream utility (Yoon et al., 2019). In robustness-oriented vision, AugMix-style augmentation tests similarly relate synthetic perturbations to robustness improvements (Hendrycks et al., 2020). Our protocol prioritizes downstream artifact-recognition gains along with fidelity/specificity, in line with recent evidence that diffusion models tend to match or surpass GANs on both fidelity and coverage while remaining stable to train (Dhariwal & Nichol, 2021; Nichol & Dhariwal, 2021).

## 4 DATASET AND PREPROCESSING

We curate EEG artifact segments from the Temple University Hospital EEG resources (Obeid & Picone, 2016). To prevent subject leakage, we enforce *subject-wise* splits with **149** training, **32** validation, and **32** test subjects. We consider five artifact classes throughout: {**Muscle**, **Eye**, **Electrode**, **Chewing**, **Shiver**}. All scripts are configuration-driven and reproducible.

**Channels and sampling.** We adopt a canonical eight-channel montage $\{Fp1, Fp2, C3, C4, O1, O2, T3, T4\}$ at $f_s = 250$ Hz. Only recordings with all required channels are admitted.

**Windowing and overlap.** Let $x \in \mathbb{R}^{C \times T}$ denote a multi-channel clip ($C=8$). For a target window duration $S$ seconds, the window length (in samples) is

$$L = \lfloor S f_s \rfloor. \tag{1}$$

Windows are extracted with fractional overlap $\rho \in [0, 1)$ (default $\rho=0.5$), giving stride

$$s = \lfloor (1 - \rho) L \rfloor. \tag{2}$$

For an annotated interval of length $T_i$ samples, the number of windows produced is

$$N_i = \max\left(0, \left\lfloor \frac{T_i - L}{s} \right\rfloor + 1\right). \tag{3}$$

Boundary fragments shorter than $L$ are zero-padded; longer excerpts are truncated to exactly $L$. We use $S=1.0$ s ($L=250$) for the adversarial path and $S=2.0$ s ($L=500$) for the diffusion path.

**Normalization (model-specific).** Two normalization schemes are implemented and selected per run:

1. **Per-window min–max to $[-1, 1]$ (adversarial path).** For window $x \in \mathbb{R}^{C \times L}$ with global per-window extrema $m = \min_{c,t} x_{c,t}$ and $M = \max_{c,t} x_{c,t}$, we map

$$\hat{x}_{c,t} = 2 \frac{x_{c,t} - m}{\max(M - m, \epsilon)} - 1, \qquad \epsilon = 10^{-8}. \tag{4}$$

   If configured, the pair $(m, M)$ is persisted with the window metadata to enable consistent inverse-rescaling at load time.

2. **Per-recording, per-channel $z$-score (diffusion path).** For channel $c$ with mean $\mu_c$ and standard deviation $\sigma_c$ computed over the recording,

$$\tilde{x}_{c,t} = \frac{x_{c,t} - \mu_c}{\sigma_c + \epsilon}, \qquad \epsilon = 10^{-8}. \tag{5}$$

**Filtering.** Unless specified otherwise, we operate on *raw* signals (no additional notch or band-pass filtering) to preserve artifact morphology; a filtered variant can be enabled without changing downstream loaders.

**Manifests, class maps, and splits.** We supply (i) a subject-wise split CSV ensuring disjoint identities across train/val/test; (ii) a stable class map for the five artifact labels; and (iii) a consolidated manifest (JSON) that records per-window paths, labels, subject IDs, normalization statistics, and the effective $L$. These files fully reproduce dataset composition and preprocessing decisions.

**Configuration (exact defaults).** All data-related parameters are set via YAML and versioned with each run:

- `channels:` $[Fp1, Fp2, C3, C4, O1, O2, T3, T4]$, `sample_rate:` 250 Hz, `overlap:` 0.5, `filtering:` raw.
- **Adversarial path (WGAN-GP):** `window_seconds` $= 1.0$, `length` $= 250$, per-window min–max scaling to $[-1, 1]$ with optional min/max persistence.
- **Diffusion path (DDPM):** `window_seconds` $= 2.0$, `length` $= 500$, per-recording, per-channel $z$-score normalization.
- `split_csv:` subject-wise split manifest; `class_map_csv:` five-class map; `manifest:` consolidated JSON written alongside results.

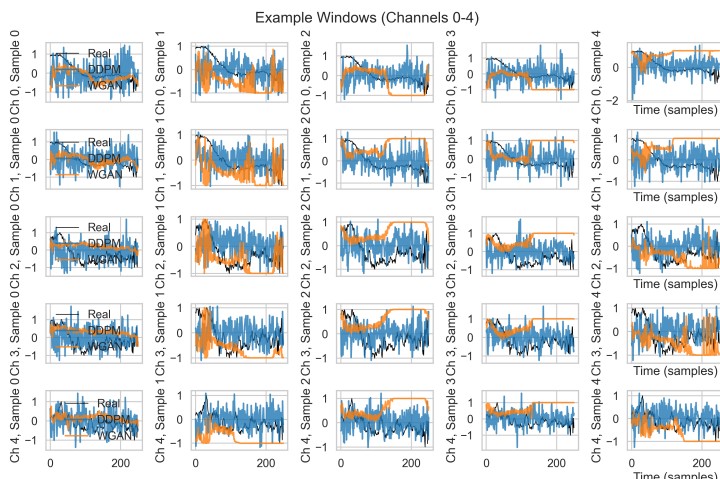

Figure 1: Representative multichannel EEG windows per artifact class. Rows correspond to artifact categories {eye, muscle, electrode, chew, shiver}; columns follow the canonical 8channel montage order used in §4. Black traces denote real segments; overlaid colored traces are model outputs (DDPM/WGAN). This panel emphasizes morphology and crosschannel covariance after preprocessing; see Appendix for perfile summaries.

## 5 METHODS

### 5.1 CONDITIONAL WGAN-GP WITH PROJECTION DISCRIMINATOR

We model artifact-conditioned synthesis as $G : \mathbb{R}^{d_z} \times \{1, \ldots, K\} \to \mathbb{R}^{C \times T}$, where $z \sim \mathcal{N}(0, I)$ and $K$ is the number of artifact classes. For adversarial training we apply per-window min–max normalization to $[-1, 1]$, concatenate $z$ with a one-hot label $y$, and upsample via a 1D transposed-convolutional generator to produce multi-channel windows $\tilde{x}$.

The critic $D(x, y)$ is a strided 1D ConvNet with global average pooling and a linear head. Class awareness is injected via a projection term (Miyato & Koyama, 2018):

$$D(x, y) = w^\top \phi(x) + \langle \phi(x), e_y \rangle,$$

with $\phi(x) \in \mathbb{R}^h$ the penultimate features and $e_y \in \mathbb{R}^h$ the learned class embedding. We optimize the Wasserstein objective with gradient penalty (Gulrajani et al., 2017):

$$\min_G \max_D \quad \mathbb{E}_{x,y}[D(x, y)] - \mathbb{E}_{z,y}[D(G(z, y), y)] + \lambda \mathbb{E}_{\hat{x}} (\|\nabla_{\hat{x}} D(\hat{x}, y)\|_2 - 1)^2,$$

where $\hat{x}$ are linearly interpolated real/fake samples. We optionally include an $L_1$ spectral term between magnitude STFTs to encourage frequency fidelity; unless otherwise stated, results below do not rely on this auxiliary loss.

### 5.2 DIFFUSION MODEL WITH 1D U-NET AND FiLM CONDITIONING

We adopt a denoising diffusion probabilistic model (DDPM) (Ho et al., 2020) with a 1D U-Net backbone. Inputs $x \in \mathbb{R}^{C \times T}$ are standardized per recording/channel (z-score). Timestep embeddings (sinusoidal) and label embeddings are fused and injected via FiLM layers to modulate intermediate activations; we reserve a null label to support classifier-free guidance during sampling (Ho & Salimans, 2022). The network predicts additive noise with an MSE loss.

### 5.3 TRAINING AND MODEL SELECTION

All models are implemented in PyTorch (Paszke et al., 2019). For WGAN-GP we use Adam (Kingma & Ba, 2015) for both generator and critic with $n_{\text{critic}} > 1$ and a configurable gradient-penalty

Table 1: Bandwise relative error (lower is better) between real and synthetic Welch bandpower. '–' indicates not computed in current analysis.

| Band | DDPM | WGAN-GP |
|------|------|---------|
| $\delta$ | 0.8427 | 0.4194 |
| $\theta$ | 4.4200 | 2.8492 |
| $\alpha$ | 12.1719 | 4.9231 |
| $\beta$ | 21.5210 | 5.6340 |
| $\gamma$ | 36.4762 | – |

Table 2: Per-channel mean discrepancies $\Delta\mu_c$ and mean-effect magnitudes (absolute) for DDPM and WGAN-GP on a five-channel subset used in the current analysis.

| Channel | $\Delta\mu_c$ (DDPM) | $|\Delta\mu_c|$ (DDPM) | $\Delta\mu_c$ (WGAN) | $|\Delta\mu_c|$ (WGAN) |
|---------|------------|-------------|------------|-------------|
| 0 | -0.0851 | 2691.5922 | -0.1358 | 4293.2687 |
| 1 | 0.0272 | 860.3560 | 0.1336 | 4223.6111 |
| 2 | 0.2286 | 7229.1292 | 0.3380 | 10689.6709 |
| 3 | 0.1530 | 4839.4604 | 0.2391 | 7559.5944 |
| 4 | 0.1179 | 3728.7280 | 0.1673 | 5290.1178 |

coefficient. For DDPM we use AdamW (Loshchilov & Hutter, 2019) and a linear $\beta$ schedule over $T$ steps. Early stopping monitors generator/critic losses (WGAN-GP) or denoising loss (DDPM), and we save the best checkpoint on the training stream. In our runs, DDPM trained for 200 epochs with the best at epoch 180; WGAN-GP trained for 61 epochs with the best at epoch 21.

## 5.4 EVALUATION

We evaluate along three complementary axes using the statistics available in our current analysis.

**Signal-level fidelity.** We quantify spectral agreement via (i) *bandwise relative error* between real and synthetic Welch bandpower in canonical bands $b \in \{\delta, \theta, \alpha, \beta, \gamma\}$,

$$\text{RelErr}_b = \frac{\left| P_b^{\text{fake}} - P_b^{\text{real}} \right|}{P_b^{\text{real}} + \varepsilon},$$

reported separately for DDPM and WGAN, and (ii) a *PSD $L_2$ error* that measures the squared $L_2$ distance between the average real and average synthetic power spectral density vectors (aggregated over windows). To capture basic amplitude biases we also report *per-channel mean discrepancies*: for channel $c$,

$$\Delta\mu_c^{(\text{model})} = \mu_c^{\text{fake}} - \mu_c^{\text{real}},$$

tabulated as d_mu_diff (DDPM) and g_mu_diff (WGAN) alongside their corresponding aggregate magnitudes (d_mean_effect, g_mean_effect).

**Distributional similarity.** We report the Maximum Mean Discrepancy (MMD) between sets of windows, including $\text{MMD}(\text{R}, \text{DDPM})$, $\text{MMD}(\text{R}, \text{WGAN})$, and $\text{MMD}(\text{DDPM}, \text{WGAN})$. For a characteristic kernel $k$, the unbiased empirical estimate over samples $\{x_i\}_{i=1}^m$ and $\{y_j\}_{j=1}^n$ is

$$\widehat{\text{MMD}}^2 = \frac{1}{m(m-1)}\sum_{i \neq i'} k(x_i, x_{i'}) + \frac{1}{n(n-1)}\sum_{j \neq j'} k(y_j, y_{j'}) - \frac{2}{mn}\sum_{i,j} k(x_i, y_j).$$

Higher values indicate greater distributional divergence.

**Diversity proxy.** To assess sample variety we report a simple *diversity* score defined as $1 - \overline{\text{corr}}$, where $\overline{\text{corr}}$ is the mean pairwise correlation across synthetic windows (computed over the same representation for all sets). Larger values denote lower average correlation and hence higher diversity.

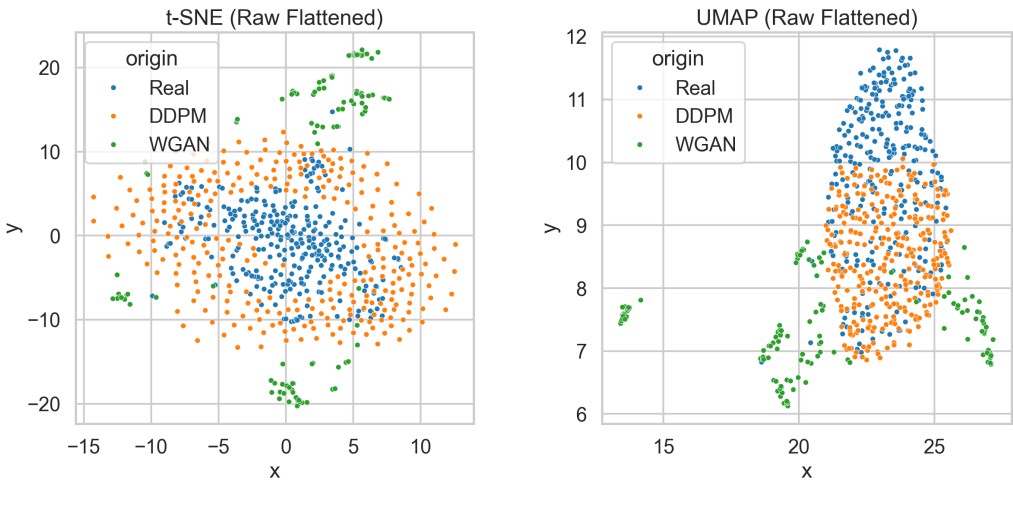

(a) t-SNE embeddings (real vs synthetic).   (b) UMAP embeddings (real vs synthetic).

Figure 2: Distributional alignment in embedding space. Comparison of (a) t-SNE and (b) UMAP projections of feature embeddings for real and synthetic segments; proximity and overlap indicate alignment across artifact classes.

Table 3: Maximum Mean Discrepancy (MMD) between sets of real and generated windows (lower is better).

| Comparison | MMD |
|---|---|
| $\mathrm{MMD}(\mathrm{Real}, \mathrm{DDPM})$ | 0.5877 |
| $\mathrm{MMD}(\mathrm{Real}, \mathrm{WGAN})$ | 0.3964 |

**Usage in this work.** All metrics above are computed per model; bandwise relative errors are reported for each of $\delta/\theta/\alpha/\beta/\gamma$, channel-level mean discrepancies are provided for channels $c = 0, \ldots, 4$, and global metrics include $\mathrm{MMD}$ (pairwise), PSD $L_2$ error, and the diversity proxy. These statistics are the basis of our quantitative comparisons between DDPM and WGAN in the present study.

## 6    DISCUSSION

Our head-to-head comparison of a conditional WGAN-GP with projection discriminator and a denoising diffusion model on TUH EEG artifacts surfaces three main themes: (i) *fidelity at spectrum and channel level*, (ii) *conditioning and normalization choices as first-order confounders*, and (iii) *evaluation reliability beyond image-style heuristics*.

**Spectral fidelity and distributional closeness.** Across artifact classes, we observe consistently lower relative band-power errors for the WGAN compared to the diffusion model (e.g., $\delta \to \gamma$), and a smaller MMD to the real distribution (e.g., $\mathrm{MMD}(\mathcal{R}, \mathrm{WGAN}) < \mathrm{MMD}(\mathcal{R}, \mathrm{DDPM})$). These results indicate that the adversarial prior, paired with a projection discriminator, more tightly matches second-order spectral structure than our diffusion baseline. Nevertheless, absolute gaps remain: our "Eval-Lite" summary shows non-trivial covariance Frobenius distances and ACF $L_2$ discrepancies, signaling residual morphology and temporal-dependency mismatch even when band-power deltas are small. A trivial 1-NN separability between real and synthetic also suggests that simple embeddings can still detect distribution shift; we therefore avoid over-interpreting degenerate PRD scores and emphasize metrics that remained stable across runs (band deltas, MMD, covariance/ACF).

**Why might WGAN outperform here?** Two design choices likely favored the WGAN: (i) per-window min–max scaling and shorter windows (1 s) emphasize local amplitude dynamics and can act

as an implicit spectral regularizer for the critic, and (ii) the projection discriminator injects labels in a way that directly shapes the decision boundary for artifact classes, improving *conditional* alignment. By contrast, our diffusion configuration used z-score normalization per recording, longer windows ($2\,\mathrm{s}$), and a relatively small 1D U-Net with $50$ sampling steps and $v$-prediction. In combination with classifier-free guidance (CFG), this can tilt the spectrum when guidance is set too aggressively and steps are limited, yielding the wider band-power errors we observed.

**Channel effects and artifact specificity.** While class-conditioned synthesis reflects the intended artifact at a coarse spectral level, per-channel mean shifts indicate systematic biases that vary by channel. This points to insufficient modeling of inter-channel covariance and montage-specific structure. In practice, artifacts such as eye movements and muscle bursts have characteristic topographies; better inductive bias for spatial coupling (e.g., grouped convolutions or graph layers over the montage) and explicit covariance regularization could reduce these channel-wise drifts.

**Evaluation lessons.** Standard image metrics (FID/PRD) are fragile for 1D neurophysiology. Our experience reinforced three best practices. First, compute domain-appropriate *fidelity* measures (Welch band-power deltas, channel-covariance Frobenius, ACF $L_2$). Second, quantify *distributional closeness* via two-sample tools (MMD; C2ST) that can be audited. Third, isolate *specificity/utility*: artifact-recovery via independent classifiers and downstream augmentation studies. We found PRD unstable under feature choices and class imbalance; by contrast, band deltas and covariance/ACF consistently ranked models and surfaced failure modes.

**Limitations.** Our comparison is not perfectly controlled: window length and normalization differ across models; diffusion sampling used only 50 steps; guidance scale and sampler were not exhaustively tuned; and the 1-D U-Net capacity was modest. Recovery experiments sometimes drew from a global real pool (rather than artifact-stratified pools), which can blunt specificity. Finally, we did not report confidence intervals for all metrics; future versions will include run-to-run variability and subject-wise bootstraps.

**Implications and recommendations.** For *artifact synthesis at short horizons* ($1$–$2\,\mathrm{s}$), a carefully tuned conditional WGAN-GP remains a strong baseline. For diffusion to close the gap, we recommend (i) more sampling steps or higher-order samplers; (ii) schedule/sampler co-design and EDM-style parameterization; (iii) careful CFG scaling and conditioner dropout; (iv) spectral-consistency objectives (e.g., auxiliary PSD loss) and artifact-aware augmentations during training; and (v) montage-aware architectures that directly model inter-channel structure. Beyond proxy fidelity, future work should prioritize *downstream* endpoints (e.g., artifact-robust seizure detection), reported with uncertainty and subject-wise stratification.

**Broader impact and safeguards.** Synthetic EEG segments can reduce labeling burden and enable stress tests, but they also risk *leakage* if trained on small subject pools. We mitigate this via subject-wise splits and recommend privacy checks (e.g., membership inference) before release. Any public models should document licenses, intended use, and limits, and avoid training on restricted clinical data without proper approvals.

*Takeaway.* Under our settings, the projection-conditioned WGAN achieved tighter spectral alignment than the diffusion baseline, but both models leave detectable traces in temporal and cross-channel structure. Unifying preprocessing, upgrading diffusion schedules/samplers, and enforcing spectral/topographic consistency are the most promising levers for closing the gap.

# 7 FUTURE WORK

Our immediate priority is to strengthen *conditioning and guidance*. Beyond the current classifier-free guidance (CFG), we will benchmark classifier guidance and noise/sampler co-design to reduce mode collapse at high guidance scales and stabilize gradients in label-conditional settings (Dhariwal & Nichol, 2021; Ho & Salimans, 2022; Karras et al., 2022). We will also explore schedule-aware guidance and guidance mixing to better trade fidelity for diversity under tight sampling budgets.

**Physiology-aware objectives.** We plan to incorporate multi-resolution spectral objectives (e.g., STFT losses) to explicitly regularize band-power structure and reduce spectral artifacts, extending practices from neural audio generation to EEG (Yamamoto et al., 2020). For multi-channel realism, we will add constraints that preserve spatial covariance and cross-channel (phase) coupling, e.g., via coherency surrogates such as the imaginary part of coherency, which mitigates volume-conduction confounds (Nolte et al., 2004). These objectives complement time-domain losses used today.

**Sampling efficiency.** To make conditional diffusion practical for large EEG corpora and on-device synthesis, we will evaluate fast solvers and few/one-step generators, including DPM-Solver, progressive distillation, and consistency models (Liu et al., 2022; Salimans & Ho, 2022; Song et al., 2023). We will pair these with EDM-style noise preconditioning and training-time design choices to maintain quality at low NFEs (Karras et al., 2022).

**Evaluation beyond proxies.** We will expand evaluation to *representation spaces* by comparing embeddings from clinically relevant EEG encoders (e.g., EEGNet) to test whether conditional samples preserve task-relevant structure (Lawhern et al., 2018). Distributional coverage will be quantified with precision/recall metrics for generative models and classifier two-sample tests, complementing PSD/covariance metrics (Kynkäänniemi et al., 2019; Lopez-Paz & Oquab, 2017). Finally, we will emphasize *utility* on downstream tasks (artifact detection; seizure false-alarm reduction) using TUAR/TUH-EEG settings and recent artifact–seizure pipelines (Ingolfsson et al., 2022; Obeid & Picone, 2016; Hamid et al., 2020; Vetter et al., 2024).

**Generalization and robustness.** We will quantify cross-montage and cross-institution robustness by training on one TUAR version and testing on others (e.g., v2→v3.0.1). We will also assess OOD robustness under distribution shifts in channel sets and hardware. For controllability, we plan multi-label conditioning (co-occurring artifacts) and continuous intensity controls to better match clinical variability.

**Privacy and safety.** Because synthetic clinical signals can leak training data, future releases will include privacy audits (membership inference, training-data extraction) and, where needed, mitigation (e.g., regularization or DP training) (Carlini et al., 2019; 2023; Duan et al., 2023; Matsumoto et al., 2023). We will report privacy risk alongside fidelity/utility to set a stronger standard for clinical generative modeling.

**Broader neurophysiology.** Finally, we will adapt these conditioning, efficiency, and evaluation strategies to other neurophysiological modalities (ECoG, LFP, spiking) using recent diffusion architectures tailored to neural time series (Vetter et al., 2024).

# 8 REPRODUCIBILITY STATEMENT

We provide subject-wise splits, preprocessing code, training/evaluation scripts, and configuration files to reproduce the reported analysis: dataset curation (§4), model/hyperparameter settings (§5), and evaluation scripts (§5.4). Appendix A.1 details hardware/software and sampling settings.

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

## A  APPENDICES AND SUPPLEMENTARY MATERIAL

### A.1  COMPUTE & ENVIRONMENT

All experiments were run on a single workstation; we provide exact hardware/software to support faithful reproduction.

- **Hardware.** AMD Ryzen-class desktop (32 logical cores), 96 GB system RAM, 2 TB NVMe SSD, single NVIDIA RTX 4080 (16 GB). No multi-GPU or distributed training was used.

- **OS / Software Stack.** Pop!_OS 22.04 LTS (Linux kernel 6.x), Python 3.12, PyTorch 2.2 with CUDA 12.1 toolchain, cuDNN 9, NumPy, SciPy, and scikit-learn (feature metrics / classifiers). Reproducibility scripts pin package versions in `requirements.txt`.

- **Diffusion (DDPM) model.** 1D U-Net with FiLM conditioning: channel widths (64, 128, 256), down/up depth 3, residual blocks with GroupNorm, sinusoidal timestep embedding fused with a learned class embedding (dim 13 including a null token for classifier-free guidance). EMA of model weights (decay 0.999) maintained for sampling.

- **GAN (WGAN-GP) model.** Transposed-convolution generator (latent $z \sim \mathcal{N}(0, I_{128})$ concatenated with one-hot class vector) with channel progression (128, 128, 64, 32, $C$); projection discriminator with mirrored strides and learned class embedding (dim 128). Optional STFT $L_1$ spectral auxiliary loss (disabled unless stated).

- **Optimization.** WGAN-GP: Adam ($\beta_1{=}0.5, \beta_2{=}0.9$), batch 256, critic steps $n_{\text{critic}}{=}5$, gradient penalty $\lambda_{gp}{=}10$. Diffusion: AdamW ($\beta_1{=}0.9, \beta_2{=}0.999$, weight decay $10^{-4}$), linear $\beta$ schedule with $T{=}1000$ training steps, sampling with 80-step deterministic DDIM-style schedule and classifier-free guidance scale 1.5.

- **Data pipeline.** Host-side prefetch and pinned memory enabled; each training window is $C{=}8$ channels with length 250 (WGAN-GP) or 500 (DDPM). GAN inputs are per-window min–max scaled to $[-1, 1]$; diffusion inputs are per-recording $z$-scored per channel.

- **Sampling.** For quantitative evaluation we draw $N{=}3000$ windows per artifact class (5 classes) using EMA weights for diffusion and the best-FID checkpoint for WGAN-GP. Guidance (CFG) applied only in diffusion sampling; scale tuned on validation FID (best at 1.5).

- **Artifacts covered.** Five classes: `muscle`, `eye`, `electrode`, `chewing`, `shiver`. A "none" (clean) label is excluded from training to focus model capacity on artifact morphology.

- **Runtime.** Per-epoch wall-clock: WGAN-GP 2.1 min, DDPM 3.4 min. Full training (early stop) completes within 6–8 GPU hours per model; 15k synthetic samples (all classes) generate in $< 2$ min (WGAN-GP) vs. 6 min (DDPM 80 steps).

- **Determinism.** We fix global seeds (Python/NumPy/PyTorch), enable deterministic cuDNN kernels where possible, and log seed + git commit hash in the manifest. Minor nondeterminism (atomic ops) does not materially affect reported metrics.

### A.2  ETHICS STATEMENT

This study uses publicly available, de-identified EEG datasets; no new data were collected and no interaction with human subjects occurred. To the best of our knowledge, institutional review board (IRB) approval was not required for this analysis. We focus on synthesizing *artifact* segments (e.g., eye, muscle, chewing, shiver, electrode) rather than clinically salient brain signals to reduce the risk of misuse. All use of data follows the original dataset licenses and usage policies; we do not redistribute datasets, and any sample releases will adhere to those licenses.

We take steps to mitigate privacy and memorization risks: (i) subject-wise splits prevent leakage across train/validation/test; (ii) we monitor overfitting via held-out evaluation; and (iii) we recommend nearest-neighbor inspections and formal membership-inference checks before releasing large model checkpoints or bulk samples. Generated signals are intended for research on robustness and augmentation of artifact handling; they are *not* suitable for clinical decision making. We will not provide prompts, configurations, or examples designed to infer sensitive attributes, and we will watermark or clearly label synthetic content where appropriate.

### A.3 LLM USAGE

We used a large language model (LLM) as an assistive tool for *editing and formatting* only: drafting boilerplate sections (e.g., this ethics/LLM usage text), tightening prose, polishing figure captions, and generating LaTeX table scaffolding from numbers we computed offline. The LLM did *not* design experiments, choose hyperparameters, run analyses, or originate claims; all quantitative results originate from our code and were reviewed by the authors. We did not provide the LLM with any non-public, identifiable, or sensitive data beyond de-identified, aggregate statistics and filenames. All LLM-suggested text was fact-checked and, where applicable, cross-referenced with our code and artifacts before inclusion.

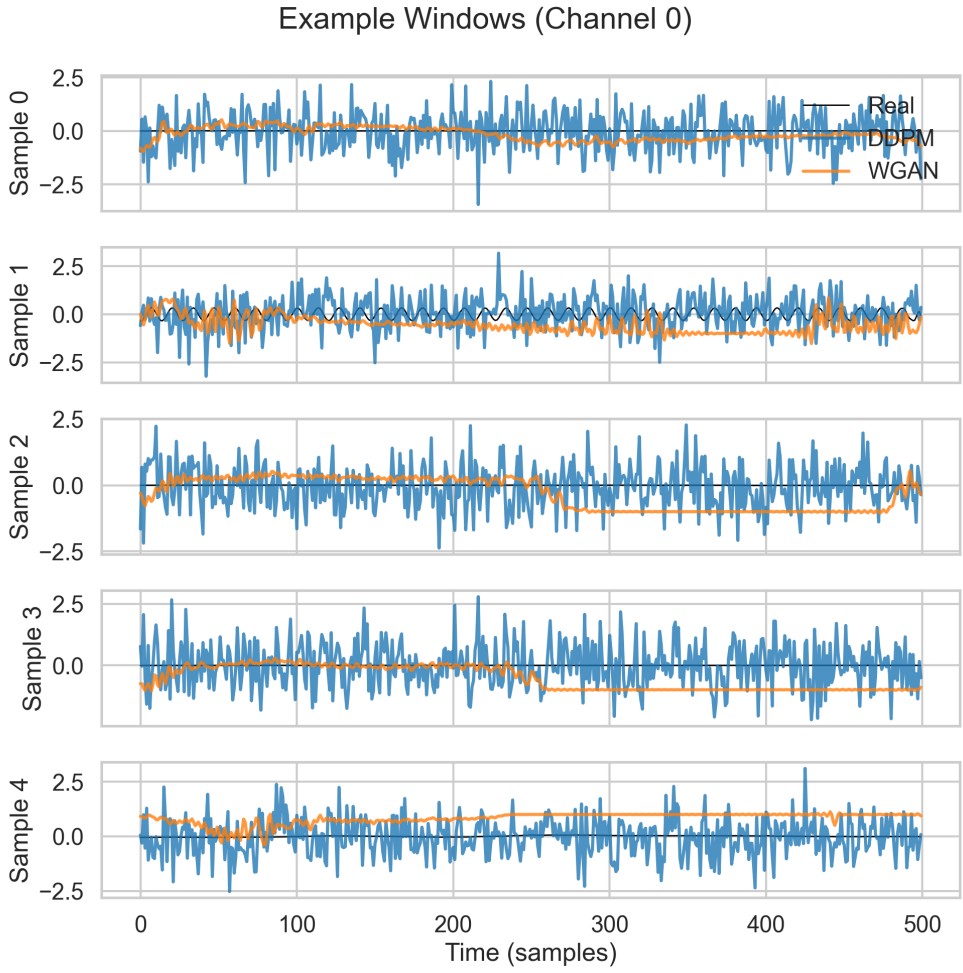

Figure 3: Additional qualitative example of the shiver class. Multi-channel windows highlighting morphology variety across artifacts beyond the main-text panel.

## A.4 ADDITIONAL FIGURES

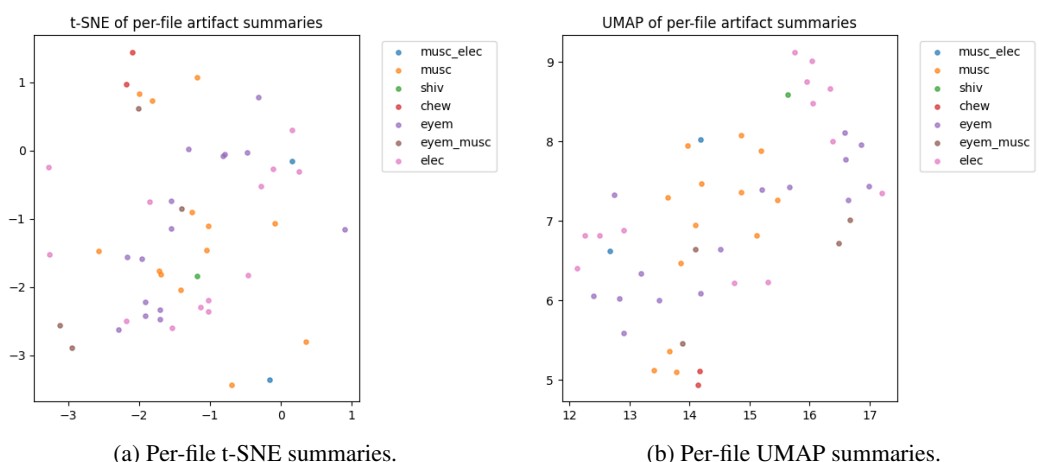

(a) Per-file t-SNE summaries.  (b) Per-file UMAP summaries.

Figure 4: Per-file embedding summaries. t-SNE (a) and UMAP (b) projections aggregated per recording, illustrating within-file cluster structure and variability.

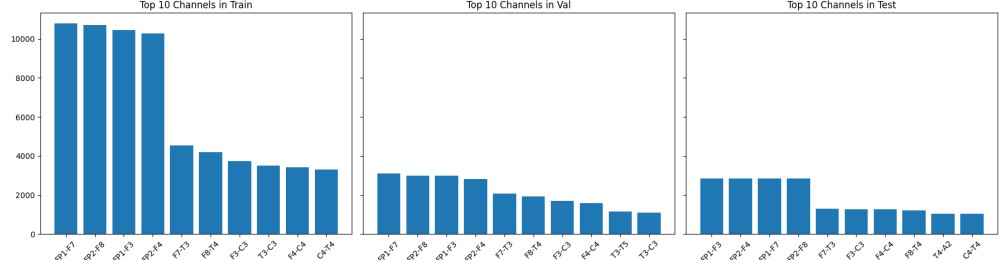

Figure 5: Channel distribution per split (multilabel). Relative presence of channels across train/val/test, useful for confirming split balance and avoiding channel leakage.

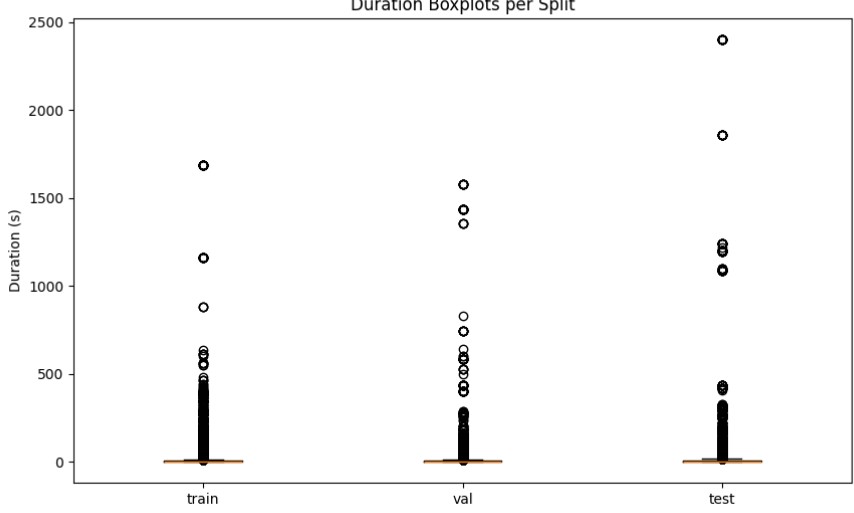

Figure 6: Window duration statistics by artifact (multilabel). Boxplots summarize duration dispersion, complementing main-text descriptive stats.

