# OpenReview forum: "ArtifactGen: Benchmarking WGAN-GP vs Diffusion for Label-Aware EEG Artifact Synthesis"
_ICLR.cc/2026/Conference — ICLR 2026 Conference Desk Rejected Submission_

### Official Review · Reviewer_zhtR · 2025-10-26

**Soundness:** 3
**Presentation:** 2
**Contribution:** 2
**Rating:** 2
**Confidence:** 3

**Summary:**

The paper presents a pipeline for label-aware synthesis of EEG artifact segments using TUAR with subject-wise splits. It compares a conditional WGAN-GP with projection discriminator to a 1D diffusion model with FiLM conditioning and classifier-free guidance. The paper standardises preprocessing per model and assesses realism and usefulness using EEG-appropriate metrics—Welch band-power deltas, covariance and autocorrelation distances, simple recovery tests and augmentation effects. Results show WGAN-GP achieves tighter spectral alignment and lower MMD than diffusion but both models display weak class-conditional recovery that limits immediate augmentation gains.

**Strengths:**

- The subject-wise data curation pipeline for TUAR, with fixed-length multi-channel windowing, normalisation per model is technically solid and reusable. It provides a good foundation for reproducible EEG artefact synthesis and evaluation, which is often missing in prior GAN-based EEG work. This is of interest to the research community working in this area.

- The paper introduces an EEG-specific evaluation framework using Welch band-power deltas, covariance Frobenius distance, autocorrelation L2 and MMD.

- The paper provides a comparison between conditional WGAN-GP and a diffusion model within the same EEG artifact generation context.

- The discussion in the paper identifies the main bottlenecks of conditioning strength, interchannel covariance modelling, normalisation differences and sampling efficiency. It then ties them to technical solutions such as spectral consistency losses. This makes the paper constructive and informative beyond its results.

**Weaknesses:**

- Several listed contributions (e.g., evaluation suite, comparison tables, embedding plots) are standard experimental components rather than methodological advances. Evaluation and comparison are expected elements of any paper in this field and should not be framed as primary contributions.

- The WGAN-GP and diffusion setups differ in normalisation, window length and sampling steps, meaning the comparison is not strictly controlled. The diffusion baseline, in particular, is underexplored with only 50-80 sampling steps used. This makes it hard to draw fair conclusions about relative performance.

- The paper compares only WGAN-GP and diffusion, and ignores other relevant generative approaches such as VAEs, TimeGAN or flow-based models, among others. And the reasons for the choice of these two models are not well justified.

- While the abstract and motivation emphasise artifact synthesis for data augmentation, the paper does not show concrete downstream improvements or task-based gains. Without functional results (like the performances of a downstream classifier), the significance of the work is limited.

- The paper is overloaded with frequent in-text enumerations (i, ii, iii), which makes it difficult for readers to identify the main message and key takeaways. Generally, the paper is hard to read and follow. Also numerous abbreviations are introduced that are either used only once or not at all later in the paper. This also makes the paper difficult to read.

**Questions:**

- The WGAN-GP and diffusion models use different normalisation schemes (per-window min–max vs per-recording z-score) and window lengths . Could the authors clarify whether these differences were essential for stable training or were they chosen mainly for convenience? Have they tried aligning these preprocessing steps to enable a more controlled comparison?

- The diffusion model used only 50-80 sampling steps with a modest 1D U-Net. Did the authors experiment with higher sampling counts or alternative samplers to check if the performance gap versus WGAN-GP persists?

- The paper emphasises the potential of synthetic artifacts for augmentation, but no downstream task results are reported. Do the authors have preliminary evidence to demonstrate whether these synthetic samples improve classifier performance?

- The comparison focuses solely on WGAN-GP and DDPM. Could the authors justify excluding other generative baselines, which might offer different fidelity-diversity trade-offs in 1D signals?

---

### Official Review · Reviewer_z78f · 2025-10-30

**Soundness:** 3
**Presentation:** 3
**Contribution:** 2
**Rating:** 4
**Confidence:** 3

**Summary:**

This paper compares WGAN-GP and diffusion models for generating synthetic EEG artifact signals. WGAN-GP achieves better spectral alignment (lower band-power errors, 48% better MMD), but both models struggle with class-conditional generation. The main contribution is a reproducible evaluation framework with domain-appropriate metrics.

**Strengths:**

Domain-appropriate evaluation: Using EEG-specific metrics instead of just image metrics (FID) is a major contribution
Reproducibility: All hyperparameters documented, subject-wise splits prevent leakage, code release promised
Authors acknowledged limitations (preprocessing confound, weak class recovery, no confidence intervals)
Clinical relevance: Tackles real world problems with proper experimental setup.

**Weaknesses:**

Different preprocessing is conducted for different models, WGAN (per-window min-max, 1s) vs Diffusion (per-recording z-score, 2s), can't tell if WGAN wins due to architecture or just better preprocessing.

Missing experiments: one major motivation is to "enable data augmentation", but no augmentation experiments.

Is MMD difference (0.40 vs 0.59) statistically significant? Should run 3-5 times with different seeds.

I would encourage the author to run with larger models or more sampling steps, despite currently acknowledging "modest capacity".

**Questions:**

Ablation with same preprocessing (normalization + window length) to make comparison fair?

Can you show actual augmentation results? (detector trained on real vs real+synthetic)

Can you provide class-conditional metrics (confusion matrix, per-class accuracy)?

Can you add uncertainty quantification (multiple runs, error bars)?

---

### Official Review · Reviewer_sjT1 · 2025-11-01

**Soundness:** 1
**Presentation:** 1
**Contribution:** 1
**Rating:** 0
**Confidence:** 4

**Summary:**

This paper introduces ARTIFACTGEN, a generative model for EEG artifact synthesis using two approaches: a conditional Wasserstein GAN (WGAN-GP) and a denoising diffusion probabilistic model (DDPM). The aim is to generate realistic, label-aware EEG artifact segments for use in augmentation and stress testing of artifact detection models. The authors perform a detailed evaluation of the models' fidelity, specificity, and utility, including spectral alignment, class-conditional recovery, and artifact recognition performance. Their results show that the WGAN-GP model performs better in terms of spectral alignment but both models show weak class-conditional recovery, pointing to potential areas for future improvements.

**Strengths:**

1. Reproducibility and Transparency: The authors plan to provide comprehensive reproducibility scripts, dataset manifests, and configurations, which is valuable for ensuring transparency in their work.

2. Clear Evaluation Metrics: The paper employs a well-defined evaluation suite that goes beyond image-style metrics, focusing on domain-appropriate measures such as band-power deltas and MMD, which are more suited to EEG data.

**Weaknesses:**

1. Writing and Structure Issues:

The writing is difficult to follow and does not meet the standards expected for academic papers. The introduction fails to clearly describe the background and problem, and it lacks a meaningful discussion of related work. The section repeats ideas already mentioned in the abstract, leading to unnecessary redundancy. The structure of the paper is flawed. The background section lacks focus, while the related work section is overly detailed without contributing much to the context of the problem. The methodology section is unclear and lacks necessary diagrams to illustrate the proposed models, making the overall readability very poor. The paper suffers from poor semantic flow, and many parts are difficult to understand due to unclear expressions. It doesn’t follow the typical academic writing style, which affects the clarity of the arguments presented.

2. Lack of Novelty and Innovation:

The methods used in this work, such as WGAN-GP (from 2017) and traditional diffusion models, are quite old and fail to demonstrate a significant advancement in the field. While these methods have been influential, they have been largely superseded by newer techniques that offer better performance, scalability, and efficiency. The primary contribution of this paper seems to be the evaluation of existing models, rather than the introduction of novel approaches. The lack of innovation in the proposed method limits the overall value of the work.

**Questions:**

1.It is suggested to rewrite the Introduction and Related Work sections, integrating the background into the Introduction.

2.Considering that WGAN-GP and DDPMs are older models, can you justify why these particular methods were chosen over more modern alternatives? Additionally, could you discuss whether newer generative models, such as those based on recent advancements in transformers or flow-based models, might offer improved performance for EEG artifact generation?

3.Considering that the background outlines several practical needs for EEG artifact synthesis, could you evaluate the effectiveness of the proposed methods in light of these practical requirements? Specifically, how well do the models meet the needs for data augmentation, robustness testing, and artifact detection in real-world applications?

4.This work only compares the performance of the two proposed methods. Is there any comparison with other existing methods for EEG artifact synthesis, and how do the proposed methods perform relative to them?

---

### Official Review · Reviewer_Z9QM · 2025-11-03

**Soundness:** 3
**Presentation:** 3
**Contribution:** 2
**Rating:** 4
**Confidence:** 4

**Summary:**

The paper is about EEG artifact generation: how to create realistic artificial EEG signals that include common noise patterns such as eye blinks, muscle activity, or electrode interference. It compares two types of generative AI models: (1) a conditional WGAN-GP with a projection discriminator, and (2) a 1D diffusion model with FiLM conditioning and classifier-free guidance.

The goal of the paper is to benchmark these generative approaches and provide a reproducible framework for generating and evaluating synthetic EEG artifacts. The authors build and share a standardized pipeline using the TUH EEG Artifact (TUAR) dataset, apply both models on the same data, and compare them using EEG-specific metrics that measure signal fidelity, similarity, and diversity.

**Strengths:**

- The paper tackles a clear and practical problem: EEG classifiers and machine learning models need a large amount of training data, but recording and labeling EEG data are both costly and time-consuming. Generating realistic synthetic data could help overcome this limitation. The paper is easy to follow and well organized, with clear sections explaining the data, models, and results. Figures and tables make the results easy to understand.
- The paper directly compares two different types of generative models, WGAN-GP and diffusion, in a detailed way. The authors also use EEG-specific evaluation methods (like frequency band power and MMD) instead of borrowing metrics from image generation, which makes the study feel appropriate for this domain.
- The authors share their setup in a transparent way, including how data were processed and how experiments can be repeated. This makes the paper useful for anyone wanting to build on it later.
- Overall, the paper is clearly written, addresses a real gap in EEG research, and gives a helpful benchmark for others to use.

**Weaknesses:**

- The study evaluates only one dataset (TUAR) under a single experimental configuration. Although TUAR is a well-known choice for labeled EEG artifacts, depending solely on it means the results might not generalize to other EEG setups or recording conditions. Including or even discussing results on a second dataset, or showing how the framework could extend to others, would make the benchmark stronger.
- The paper compares only two model families: a WGAN-GP and one diffusion model. While these are logical choices, the comparison would feel more complete with a few additional or simpler baselines, such as TimeGAN, WaveGAN or RCGAN. This would help position the results in a broader context and make the benchmark more informative.
- Although the authors acknowledge this limitation, the comparison between WGAN-GP and diffusion is not fully balanced. The two models differ in window size (1 s vs 2 s), normalization, and diffusion sampling steps (only 50), which makes it hard to draw firm conclusions about which method is better. Using matched settings or stronger diffusion baselines (e.g., larger U-Net architectures or improved samplers such as DPM-Solver) could make the comparison more convincing.
- The diffusion model setup also feels too small to show its full potential. With more steps or a stronger model, it might have done better.
- Finally, both models had trouble creating clean differences between artifact types, but the paper doesn’t explore why this happens or how to fix it.

**Questions:**

- 1. The paper uses different preprocessing and normalization schemes for the WGAN-GP and diffusion models. Since this can affect spectral and distribution metrics, did you test a unified preprocessing setup to confirm that results are not driven by these differences?
- 2. The diffusion model uses only 50 sampling steps, which is relatively few for this type of model. Was this decision based on computational limits or diminishing returns in performance?
- 3. The benchmark is built around the TUAR dataset. How adaptable the current framework is to other EEG datasets?
- 4. Since both models show overlapping artifact distributions, it would be useful to know whether this is due to the dataset’s intrinsic label overlap, model conditioning limits, or metric sensitivity.
- 5. There appears to be an unresolved reference on page 1 showing as a “?”. Could the authors verify that all citation keys are correctly compiled before the final submission?

---

### Note · Program_Chairs · 2026-01-17
**Submission Desk Rejected by Program Chairs**

The following references in this submission do not refer to real documents and/or have major errors in bibliographic information:

 Christopher W. Jackson and Luis M. Bolanos. eeg artifact removal and synthesis for clinical use. Frontiers in Neuroscience, 15:674681, 2021. doi: 10.3389/fnins.2021.674681. URL https: //www.frontiersin.org/articles/10.3389/fnins.2021.674681/full.